# Computational modeling identifies embolic stroke of undetermined source patients with potential arrhythmic substrate

Savannah F Bifulco[1], Griffin D Scott[1], Sakher Sarairah[2], Zeinab Birjandian[2,3], Caroline H Roney[4], Steven A Niederer[4], Christian Mahnkopf[5], Peter Kuhnlein[5], Marcel Mitlacher[5], David Tirschwell[3], WT Longstreth[3,6], Nazem Akoum[2]*, Patrick M Boyle[1,7,8]*

[1]Department of Bioengineering, University of Washington, Seattle, United States; [2]Division of Cardiology, University of Washington, Seattle, United States; [3]Department of Neurology, University of Washington, Seattle, United States; [4]School of Biomedical Engineering and Imaging Sciences, King's College London, London, United Kingdom; [5]Department of Cardiology, Klinikum Coburg, Coburg, Germany; [6]Department of Epidemiology, University of Washington, Seattle, United States; [7]Center for Cardiovascular Biology, University of Washington, Seattle, United States; [8]Institute for Stem Cell and Regenerative Medicine, University of Washington, Seattle, United States

*For correspondence:
nakoum@cardiology.washington.edu (NA);
pmjboyle@uw.edu (PMB)

Competing interests: The authors declare that no competing interests exist.

**Abstract** Cardiac magnetic resonance imaging (MRI) has revealed fibrosis in embolic stroke of undetermined source (ESUS) patients comparable to levels seen in atrial fibrillation (AFib). We used computational modeling to understand the absence of arrhythmia in ESUS despite the presence of putatively pro-arrhythmic fibrosis. MRI-based atrial models were reconstructed for 45 ESUS and 45 AFib patients. The fibrotic substrate's arrhythmogenic capacity in each patient was assessed computationally. Reentrant drivers were induced in 24/45 (53%) ESUS and 22/45 (49%) AFib models. Inducible models had more fibrosis (16.7 ± 5.45%) than non-inducible models (11.07 ± 3.61%; p<0.0001); however, inducible subsets of ESUS and AFib models had similar fibrosis levels (p=0.90), meaning that the *intrinsic pro-arrhythmic substrate properties* of fibrosis in ESUS and AFib are indistinguishable. This suggests that some ESUS patients have latent pre-clinical fibrotic substrate that could be a future source of arrhythmogenicity. Thus, our work prompts the hypothesis that ESUS patients with fibrotic atria are spared from AFib due to an absence of arrhythmia triggers.

## Introduction

Atrial fibrillation (AFib) is the most common cardiac arrhythmia, affecting 1–2% of the world's population and significantly contributing to worldwide morbidity and mortality (*Andrade et al., 2014*). The primary source of AFib-related mortality is stroke, with around 20% of all ischemic strokes occurring in AFib patients (*Andrade et al., 2014*). Sub-clinical AFib (i.e., transient, asymptomatic AFib) is implicated as a potential cause of embolic stroke of undetermined source (ESUS), and the current course of clinical care following ESUS is to look for evidence of AFib via an external monitor, an implanted loop recorder, or other forms of wearable monitoring devices. If AFib is diagnosed, treatment with oral anticoagulants is started to mitigate the possibility of recurrent stroke (*Israel et al.,*

**eLife digest** The heart usually beats with a regular rhythm to pump the blood that carries oxygen and nutrients to different organs. Sometimes, alterations in the heart's rhythm known as arrhythmias can occur. Atrial fibrillation, also called AFib, is a type of arrhythmia in which the heart beats rapidly and irregularly, causing abnormal blood-flow that can lead to the formation of blood clots. If one of these blood clots travels to the brain, it can block a blood vessel, causing a stroke. However, many strokes occur without any evidence of AFib.

One subset of strokes that are not associated with AFib are embolic strokes of undetermined source (ESUS), which account for 25% of all strokes. By definition ESUS and AFib do not occur together, but both are associated with similar elevated levels of disease-related remodeling (i.e., fibrosis) in the heart tissue, which appears when the heart is injured. Fibrosis impairs the heart's normal electrical activity.

Bifulco et al. wanted to determine whether there is some fundamental difference in fibrosis between people with AFib and those who have had an ESUS event. To do this, they used a computational approach to model the geometries and patterns of fibrosis of the hearts of 45 ESUS patients and 45 patients with AFib, essentially producing a virtual version of each patient's heart. Bifulco et al. then applied a virtual pace-maker (working in overdrive mode) to each heart model to determine whether electrical inputs that can lead to AFib had different effects on ESUS and AFib patients.

The results showed that the electrical inputs had similar effects in all of the heart models. This led Bifulco et al. to conclude that ESUS and AFib patients have indistinguishable patterns of fibrosis. The key difference is that ESUS patients are missing the trigger to initiate the fibrillation process – if atrial fibrosis is the proverbial tinderbox, these triggers are the spark needed to ignite a fire.

Further research, including confirmation of Bifulco et al.'s findings in live patients, will be needed to confirm the hypothesis that ESUS patients lack AFib primarily due to an absence of triggers. If this is indeed the case, these findings may make it easier to identify ESUS patients at higher risk for AFib or further strokes. Additionally, a better understanding of fibrosis as a link between stroke and AFib will help clinicians provide better, more personalized treatments, for example guiding whether a patient should take blood thinners or undergo more rigorous cardiac monitoring.

2017). Clinical studies have shown that AFib has been detected in only 30% of patients with long-term rhythm monitoring (*Brachmann et al., 2016*). This creates a frustrating problem for clinicians: in the wake of ESUS events, it is impossible to know which individuals should be treated as high-risk for AFib and therefore monitored accordingly.

Recent evidence from clinical studies suggests that the left atrial fibrosis burden measured by late gadolinium enhanced-magnetic resonance imaging (LGE-MRI) is as high in ESUS patients as in AFib patients without stroke (*Tandon et al., 2019*). This finding supports the hypothesis that atrial fibrosis is an element of the causal pathway for stroke, through an atrial cardiopathy, and independent of AFib. The absence of AFib despite the presence of a fibrotic substrate is intriguing and one potential explanation is that ESUS patients have pro-arrhythmic fibrotic substrate but lack the triggers needed to initiate arrhythmia. Another potential explanation is that the fibrosis present in ESUS patients is not pro-arrhythmic. Patient-derived computational modeling of atrial arrhythmias is uniquely poised to test these hypotheses. Previously, personalized atrial models have been used to assess arrhythmo-genic propensity of fibrotic substrate and predict AFib ablation targets (*Zahid et al., 2016a*; *Boyle et al., 2019*). Applying the same approach, we can use computational models to predict if, in the presence of appropriate triggers, fibrotic remodeling in ESUS has the fundamental capacity to harbor reentrant arrhythmic activity.

Thus, we present a large-scale computational study to ascertain whether the fibrotic substrate with the potential to perpetuate AFib-sustaining reentrant drivers (RDs) exists in ESUS. Our hypothesis is that a pre-clinical AFib substrate, attributed to a pattern of fibrotic atrial remodeling that is conducive to RD perpetuation, exists in ESUS patients. By conducting simulations in models derived from LGE-MRI, we can begin to understand potential pro-arrhythmic properties of atrial fibrosis in

ESUS patients. The study thus provides insights on the role of atrial fibrosis as a pathophysiological nexus between AFib and stroke.

## Results

### Patient characteristics

Ninety patient-derived models were included in our analysis: 45 post-stroke ESUS and 45 pre-ablation AFib patients. Demographic information about both patient groups is provided in *Table 1*. There was a significant difference in left atrial (LA) surface area; however, the potential importance of this feature for interpreting our findings is offset by the lack of difference in LA volume index (a more commonly used measurement of normalized LA surface area), which suggests that higher LA surface area in those with AFib is a consequence of higher body mass index (BMI). LA fibrosis burden was not significantly different between ESUS (13.6 ± 6.2%) and AFib patients (14.2 ± 4.5%) (p=0.91), consistent with previous findings (*Tandon et al., 2019*).

### Induction of arrhythmia and fibrosis quantification in patient-derived models

Personalized LA bilayer models were generated for all ESUS and AFib patients. Examples of physiological detail incorporated in models can be seen in *Figure 1*, including patient-specific patterns of fibrotic remodeling, realistic atrial fiber orientations, and locations of electric pacing sites; further detail can be found in the Materials and methods section and *Figure 1—figure supplement 1*. Rapid electric stimulation caused RD-sustained arrhythmia in 22 of 45 AFib models (48.8%) and 24 of 45 ESUS models (53.3%). Thus, the capability of the fibrotic substrate to sustain RDs was not significantly different between the two groups (p=0.83, *Figure 2A*). ESUS and AFib models were then sorted by amount of global LA fibrosis and arranged into quartiles. For five models (21.7%) in the first quartile (fibrosis < 9.75%), six models (28.5%) in the second quartile (fibrosis < 12.6%), 15 (62.5%) models in the third quartile (fibrosis < 17. 5%), and 20 (91.0%) models in the top quartile, simulations revealed at least one pacing site for which stimulation produced an episode of RD-sustained arrhythmia (*Figure 2B*).

To explore potential pro-arrhythmic substrate properties in ESUS and AFib, we analyzed fibrosis burden in the sub-groups of each cohort in which RD-sustained arrhythmias were inducible and non-inducible (*Figure 3A*). Fibrosis burden was not significantly different between inducible ESUS and AFib models with (*Figure 3A*; p=0.90, confidence interval; CI: [−3.4, 4.1]) or without induced reentry (*Figure 3A*; p=1, CI: [−2.1, 2.4]). However, when fibrosis burdens for inducible and non-inducible models were aggregated across ESUS and AFib groups, a significant difference was evident (*Figure 3A*; 16.7 ± 5.46% vs. 11.07 ± 3.61%; p<0.0001, CI: [3.4, 7.6]).

**Table 1.** Patient characteristics in ESUS and AFib groups.

|  | ESUS (N = 45) | AFib (N = 45) | p value |
| --- | --- | --- | --- |
| Age, years | 60 ± 16 | 62 ± 12 | 0.504 |
| Female, % | 44.0% | 32.8% | 0.275 |
| BMI, kg/m$^2$ | 27.6 ± 4.3 | 29.5 ± 5.9 | 0.08 |
| CHA$_2$DVASc score | 2.0 | 1.9 | 0.345 |
| CHF, n | 14.3% | 18.4% | 0.599 |
| Hypertension, n | 68.5% | 61.2% | 0.468 |
| Diabetes mellitus, n | 20.4% | 12.2% | 0.292 |
| CAD, n | 18.4% | 18.4% | 1.000 |
| Smoking, n | 32% | 28% | 0.679 |
| LA fibrosis, % | 13.6 ± 6.2% | 14.2 ± 4.5% | 0.91 |
| LA surface area, cm$^2$ | 109 ± 26 | 134 ± 40 | 0.0007 |
| LA volume index, mL/m$^2$ | 60 ± 29 | 57 ± 26 | 0.607 |

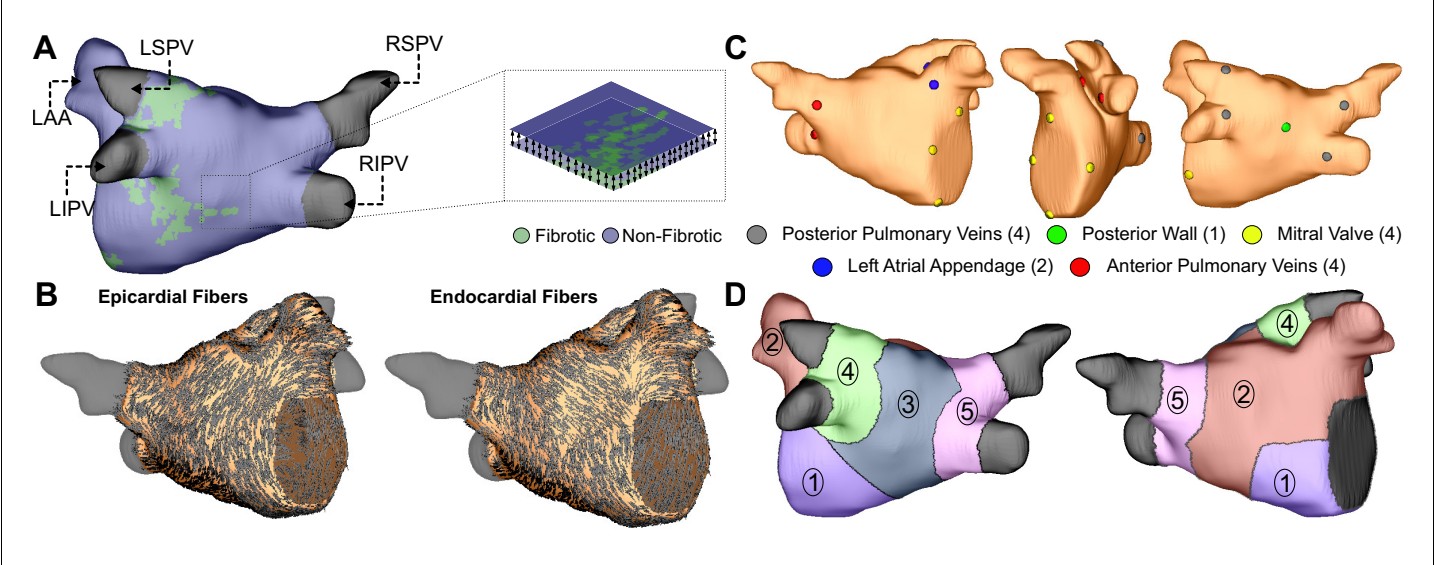

**Figure 1.** Model generation. (**A**) Reconstruction of LA geometry with anatomical features labeled (RIPV/RSPV/LIPV/LSPV, right/left inferior/superior pulmonary veins; LAA, LA appendage). The LA is modeled as a bilayer comprising nested endocardial and epicardial shells linked in both fibrotic and non-fibrotic regions by 1D linear elements. (**B**) LA fiber orientations for the endocardium and epicardium, mapped from human atlas geometry as described in Materials and methods. (**C**) AFib trigger sites as pacing sites (posterior/anterior LIPV, LSPV, RSPV, RIPV, LAA base, mitral valve annulus, and posterior wall). (**D**) Regions of the LA generated as described in methods: (*Andrade et al., 2014*) atrial floor, (*Israel et al., 2017*) anterior wall and LAA, (*Brachmann et al., 2016*) posterior wall, (*Tandon et al., 2019*) left PVs, and (*Zahid et al., 2016a*) right PVs.
The online version of this article includes the following figure supplement(s) for figure 1:

**Figure supplement 1.** LA subdivision scheme.

We also investigated potential differences in the number of unique RDs and the number of AFib triggers that induced reentry in each patient-derived model. In all 46 inducible models, the median number of unique RDs was one for both AFib and ESUS models (AFib range: 1–5; ESUS range: 1–8). There was no significant difference in the number of unique reentrant morphologies per model between the two groups (*Figure 3B*; p=0.83, CI: [−7.7 × 10⁻⁵, 8.8 × 10⁻⁵]). The number of unique RDs was positively correlated with atrial fibrosis burden (*Figure 3C*; R = 0.63, p<0.0001). For all RD-inducible cases, the median number of stimulation sites from which rapid pacing led to RD formation was three for AFib and two for ESUS models (range for both groups: 1–9). No significant difference was found in the number of pacing sites that induced reentry per model between AFib and ESUS (*Figure 3D*; p=0.79, CI: [−2.0, 1.0]). The number of RD inducing pacing sites was also significantly correlated with fibrosis burden (*Figure 3E*; R = 0.62, p<0.0001).

In addition to its role as part of the substrate for reentrant arrhythmia, fibrosis may directly lead to increased AFib trigger incidence via calcium dysregulation leading to localized to regions of depolarized resting potential (*Gouvêa de Barros et al., 2015*; *Alonso et al., 2016*). To investigate whether cohort-scale differences in this intrinsic pro-trigger property of fibrosis may explain the lack of arrhythmia in ESUS, we ran additional simulations to assess the total extent of abnormally depolarized tissue (see Materials and methods for definition) in ESUS and AFib models; notably, these values are distinct from total fibrosis burdens, since non-fibrotic tissue can be pulled to a more positive resting potential via electrotonic coupling. We found no significant difference in these values between ESUS and AFib models (*Figure 3F*; p=0.32; CI: [−0.007, 0.019]); if anything, there was a trend toward *more* trigger-prone tissue in ESUS. These findings provide preliminary evidence against the notion that the lack of arrhythmia in ESUS might be due to lower rates of fibrosis-related ectopic pacemaking.

## Arrhythmia dynamics

Analysis of simulated reentry episodes revealed no qualitative differences in arrhythmia dynamics between AFib and ESUS models. *Figure 4* shows examples of RD-perpetuated in silico arrhythmia

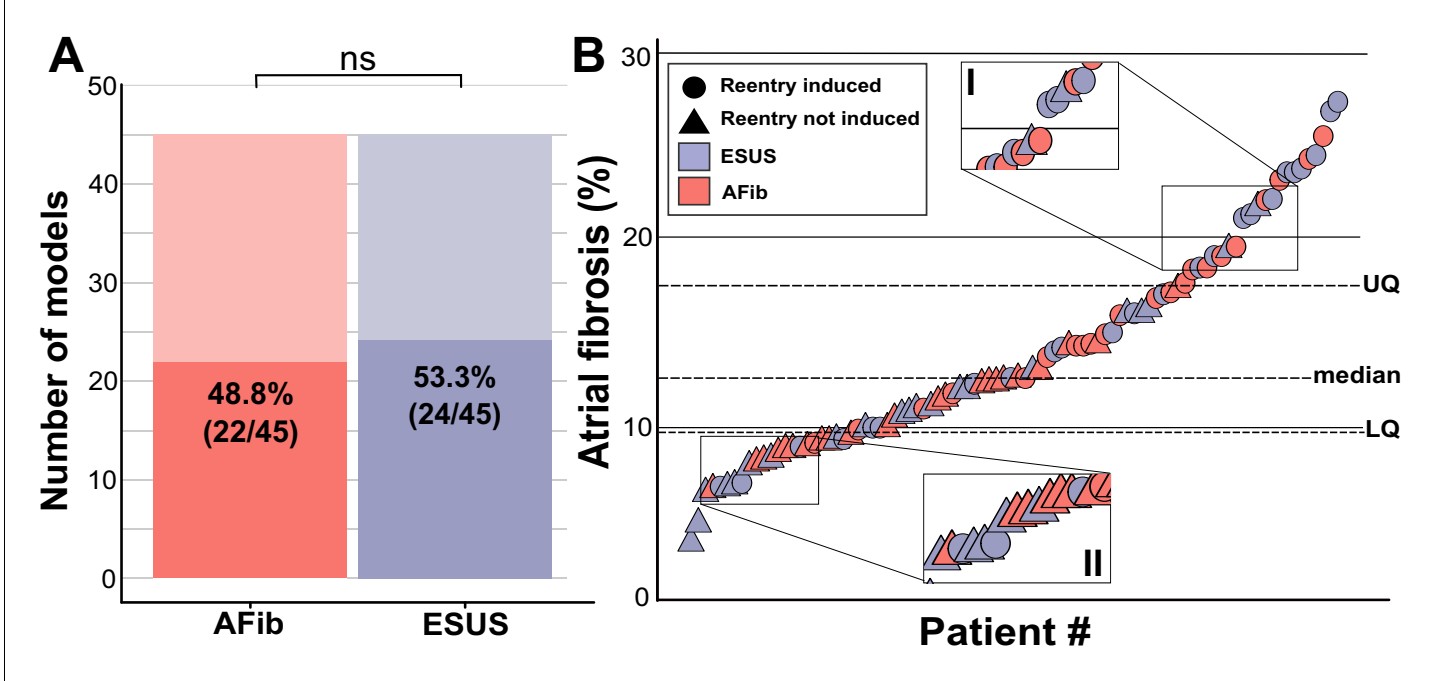

**Figure 2.** Summary of patient-derived model fibrosis with respect to RD inducibility. (**A**) Histogram of AFib (22/45) and ESUS (24/45) inducible patients. Inducibility was not significantly different by χ² test. (**B**) Patients with ESUS and AFib arranged by percentage of LA fibrosis. Dotted lines indicate the quartiles of fibrosis observed for all 90 patient-derived models. Circles are indicative of stable reentry observed in the model from at least one pacing site after in silico pacing protocol. Triangles indicate no RDs after pacing from all 15 pacing sites independently. Cases that lacked RDs despite high fibrosis (inset I) or were inducible despite low fibrosis (inset II) are highlighted.

The online version of this article includes the following source data for figure 2:

**Source data 1.** Spreadsheet including source data underlying *Figure 2*.

and instances where stimulation failed to induce reentry for both groups. Of note, this figure highlights two inducible low-fibrosis ESUS models (*Figure 4A*: 6.9% fibrosis, RD near the LIPV; *Figure 4B*: 10.0% fibrosis, RD on the atrial floor) and a non-inducible high-fibrosis ESUS model (*Figure 4C*: 16% fibrosis). In the latter case, dense fibrosis on the posterior wall resulted in conduction block as indicated.

*Figure 4D,E* present examples of RD-driven arrhythmia in AFib models (9.9% and 13.7% fibrosis, respectively), and *Figure 4F* shows an AFib model (11.6% fibrosis) in which reentry was not induced due to wavefront collision in the posterior wall region. Overall, this shows that both ESUS and AFib models exhibited activation patterns consistent with previous definitions of RD-driven arrhythmia; examples of inducible low-fibrosis and non-inducible high-fibrosis models emphasize that fibrosis burden alone is an insufficient predictor for a potential arrhythmic substrate.

## Properties of RD localization

As described in Materials and methods, each model was automatically subdivided into five anatomical regions (see schematic illustrations in *Figure 1D*, *Figure 1—figure supplement 1*), and region-wise inducibility score analysis (IdS, as described in Materials and methods) was used to gauge likelihood of RD induction in response to rapid electrical stimulation from different locations in the LA. While ESUS and AFib models had a statistically similar pattern of inducibility rates (p=0.45, by χ² test), stimulation from the posterior wall was approximately two times more likely to induce RDs in AFib models than ESUS models (*Figure 5A*, IdS = 6 vs. IdS = 3). In other words, with all other factors held equal, our simulations suggest that triggered activity in the posterior wall may be more likely to initiate reentrant arrhythmia in AFib patients compared to ESUS patients. The same IdS values plotted in *Figure 5A* were mapped onto representative LA models to facilitate visual comparison of regional sensitivity to rapid pacing (*Figure 5B*). Next, we considered the number of unique RD localization sites in each LA region across the different model groups (i.e., AFib vs. ESUS). The LPV region

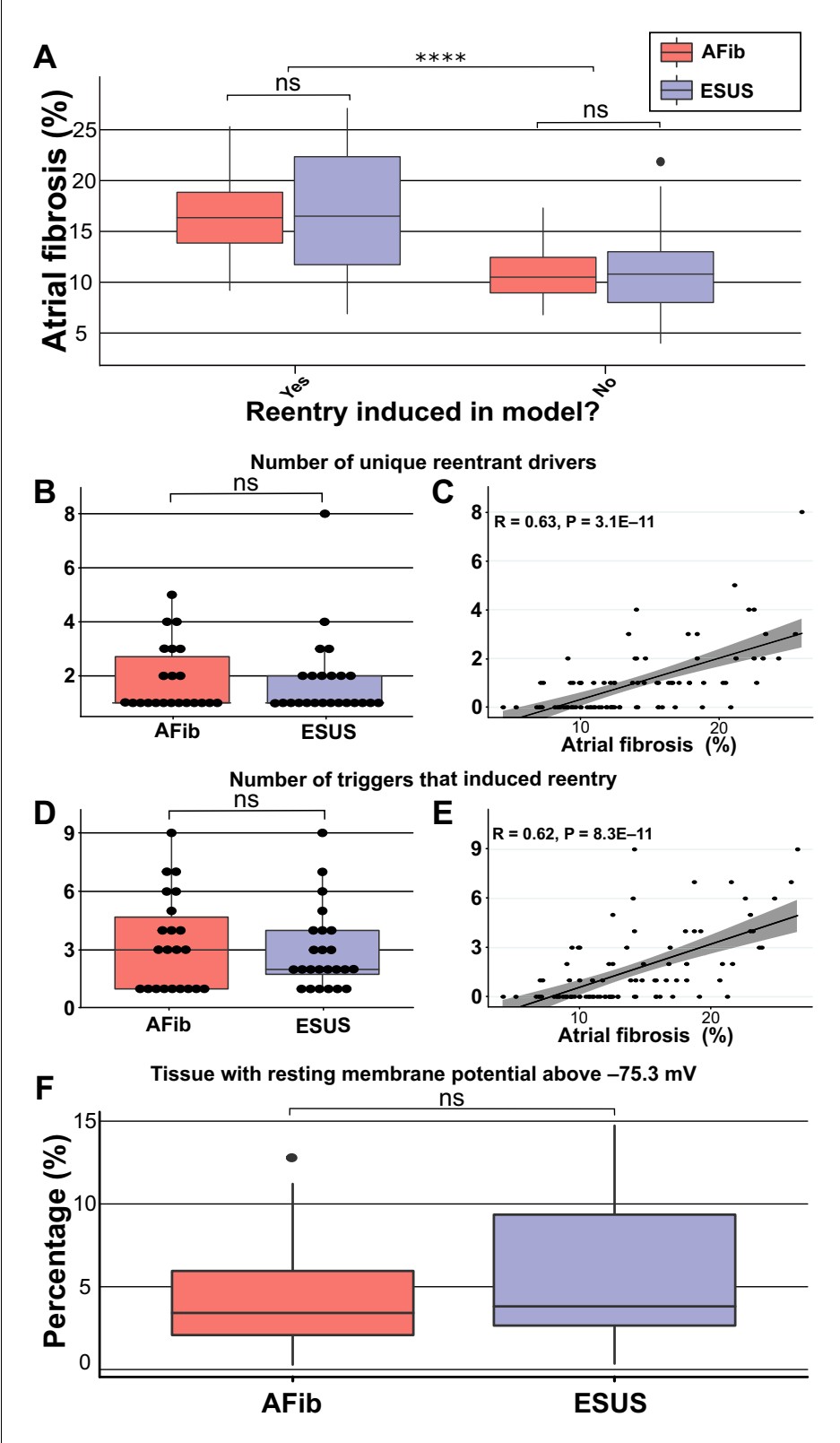

**Figure 3.** Summary of RD characteristics between ESUS and AFib models. (**A**) Boxplot of fibrosis percentage in ESUS and AFib models where reentry was induced (ESUS: N = 24, IQR = 10.6; AFib: N = 22, IQR = 5) and where reentry was not induced (ESUS: N = 21, IQR = 5; AFib N = 23, IQR = 3.5). Across ESUS and AFib models, fibrosis burden for RD-inducible and RD non-inducible models was significantly different (p<0.0001). (**B**) Boxplot of number of unique reentrant morphologies elicited by all 15 pacing sites (p=0.83). (**C**) Correlation plot of fibrosis vs. number of RDs (R = 0.63, p<0.0001).
*Figure 3 continued on next page*

Figure 3 continued

(D) Boxplot of the number of pacing sites which induced reentry (p=0.79). (E) Correlation plot of fibrosis vs. number of pacing sites that induced reentry (R = 0.62, p<0.0001). (F) Boxplot depicting percentage of tissue with significantly depolarized tissue (>95th percentile) between ESUS and AFib models after reaching steady state. p=0.32; CI: [–0.007, 0.019].

The online version of this article includes the following source data for figure 3:

**Source data 1.** Spreadsheet including source data underlying *Figure 3*.

was most likely to harbor RDs in the AFib cohort (*Figure 5C*, N = 17). In the ESUS cohort, the anterior wall was the most likely region to contain an RD (*Figure 5D*, N = 16). The association between the type of model (ESUS vs. AFib) and RD localization was not significant (p=0.13, χ² test).

## Properties of pro-RD fibrosis

To further examine trends in RD inducibility and localization observed in *Figure 5*, we also carried out region-wise analysis of fibrosis spatial pattern. In a previous study (*Zahid et al., 2016a*), machine learning was used to quantitatively characterize the fibrosis distribution in regions that most frequently harbored RD organizing centers (i.e., high local fibrosis density *and* entropy). As described in Materials and methods, the distribution of pro-RD tissue regions was determined for each inducible model; these values were then subdivided in the five-region schematic (*Figure 6A*). This analysis

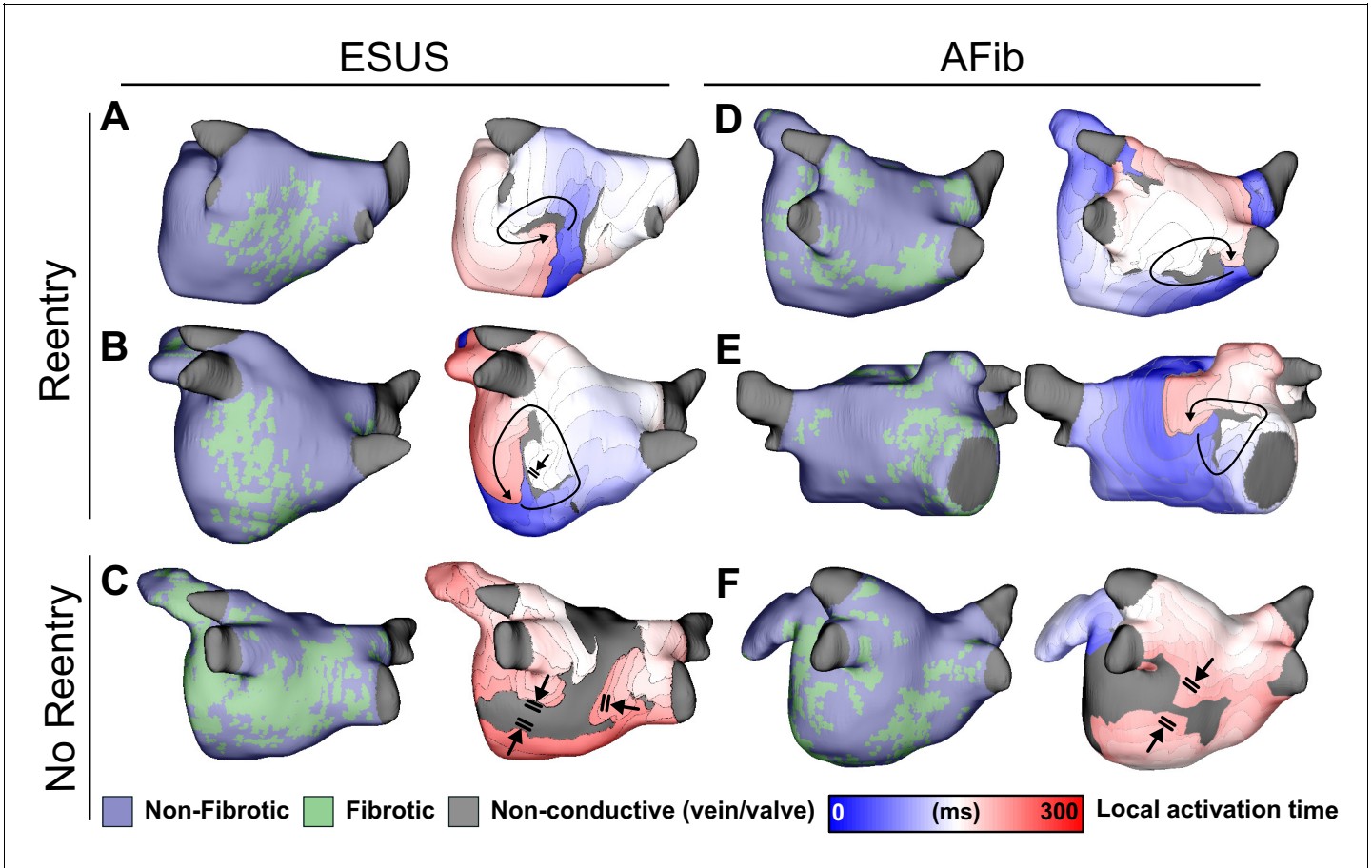

**Figure 4.** Maps of fibrotic tissue distribution (left) and activation time (right) for ESUS and AFib models in which pacing succeeded (rows 1–2) or failed (row 3) to induce RD-driven arrhythmia. Black arrows indicate directions of wavefront propagation in RDs. Double lines indicate sites of conduction block. Black-shaded regions in activation maps indicate locations where activation did not occur during the analysis interval. (A) ESUS model with 6.9% fibrosis and reentry inferior to LIPV. (B) ESUS model with 10.0% fibrosis and reentry on the atrial floor. (C) ESUS model with 16% fibrosis with wavefront termination through fibrosis on posterior wall. (D) AFib model with 9.9% fibrosis and reentry observed adjacent to RIPV on posterior wall. (E) AFib model with 13.7% fibrosis and reentry observed on the anterior wall. (F) AFib model with 11.6% fibrosis with wavefront collision on posterior wall.

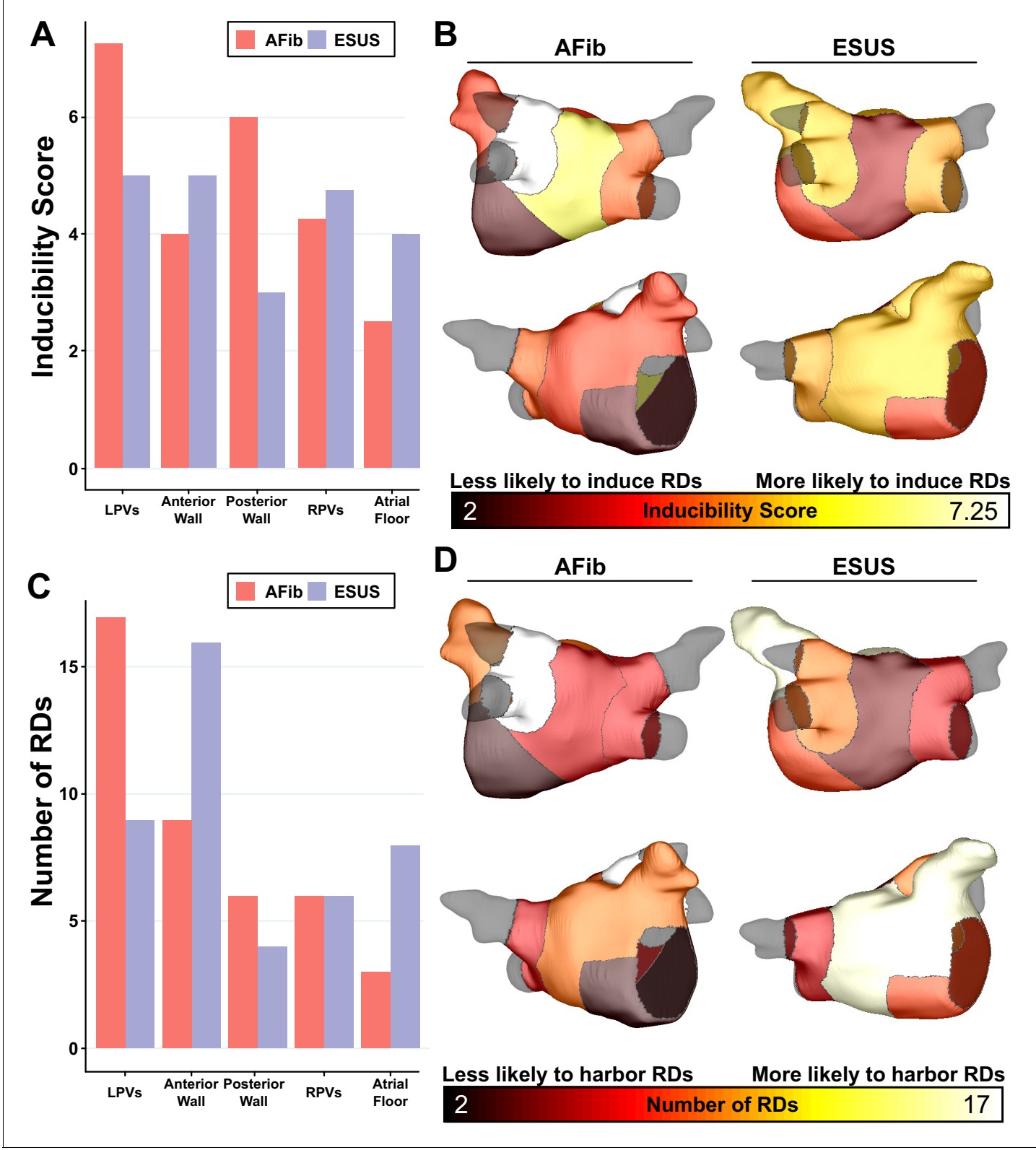

**Figure 5.** Summary of IdS and RD localization characteristics. (**A**) Region-wise IdS for both ESUS and AFib LA models. (**B**) Heat map of the regions in which triggers are most likely to induce arrhythmias depicted as representative ESUS and AFib models. (**C**) Histogram of RDs across all AFib and ESUS models binned by localization to specific LA regions. (**D**) Heat map of regions in which RDs are most likely to localize depicted as representative ESUS and AFib models.

*Figure 5 continued on next page*

*Figure 5 continued*

The online version of this article includes the following source data for figure 5:

**Source data 1.** Spreadsheet including source data underlying *Figure 5*.

revealed that the extent of pro-RD tissue in the LPV region was significantly higher in AFib compared to ESUS models (*Figure 6B*; p=0.01); this was consistent with our findings on regional inducibility, and RD localization as shown in *Figure 5*. Likewise, for regions that were associated with greater inducibility and RD localization in ESUS models (anterior wall+LAA, atrial floor), there was a trend toward a larger extent of pro-RD tissue compared to AFib models, but these did not reach the level of significance.

## Discussion

This study used a novel computational modeling approach with stimulus locations chosen based on clinically observed AFib triggers to shed new light on the role of the fibrotic atrial substrate in the potential for initiation and perpetuation of reentry in ESUS patients. In models reconstructed from 45 post-stroke ESUS and 45 pre-ablation AFib patients, we showed that the AFib and ESUS groups did not differ significantly in (1) the propensity of the fibrotic substrate to sustain RDs in response to simulated burst pacing; (2) the LA fibrotic burden of RD-inducible models or RD-free models; and (3) the RD localization or the region-wise inducibility. One noteworthy difference was that the extent of tissue in the LPV region with a pro-RD fibrosis spatial pattern was greater in AFib vs. ESUS models. This is the first study to use computational modeling and simulation to assess potential pro-arrhythmic capacity of LA fibrosis in ESUS patients. Moreover, to the best of our knowledge, this is the largest cohort ever studied via computational analysis of atrial electrophysiology in models derived from LGE-MRI, exceeding the number of patient-specific models (N = 50) in the former largest study (*Roney et al., 2020*) by a factor of ≈1.8.

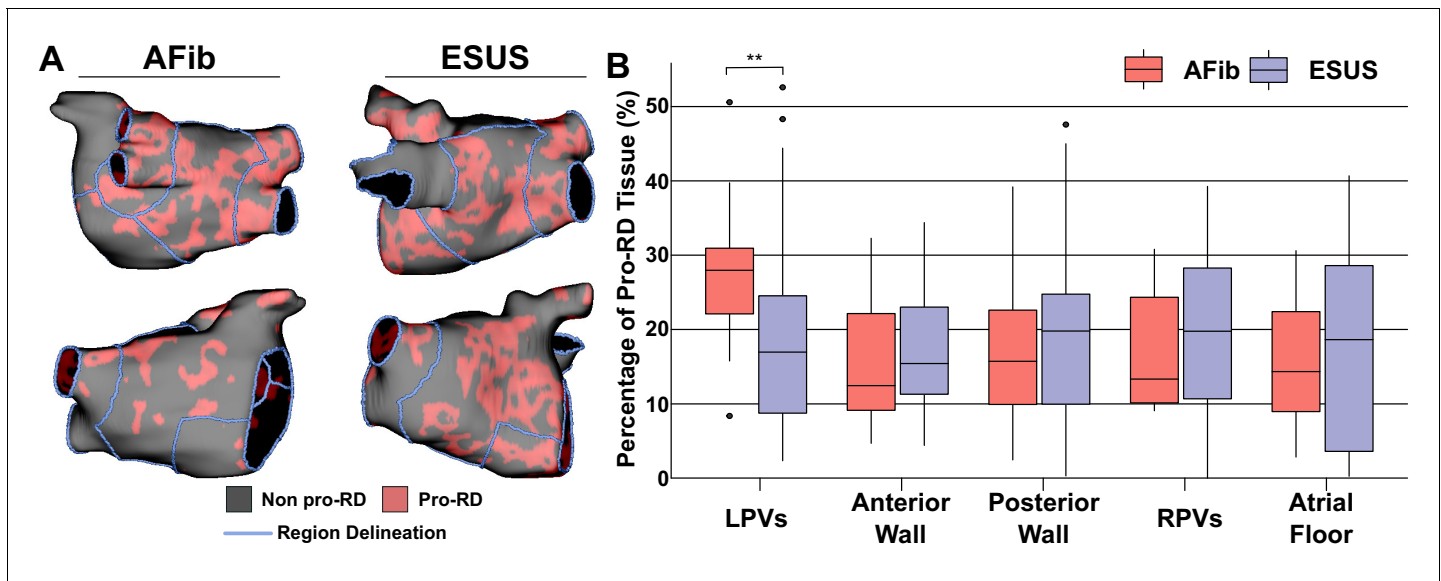

**Figure 6.** Summary of the region-by-region extent of tissue with a spatial fibrosis pattern (as characterized by local density and entropy) associated with RD localization (i.e., pro-RD tissue). (**A**) Maps of pro-RD tissue for representative AFib and ESUS cases, including boundaries between regions. (**B**) Region-wise extent of pro-RD tissue in inducible AFib and ESUS models, depicted as boxplots (**p<0.01, Wilcoxon rank-sum test; CI: [–0.149, –0.021]). The online version of this article includes the following source data for figure 6:

**Source data 1.** Spreadsheet including source data underlying *Figure 6*.

## Inducibility of reentry and fibrosis quantification in patient-derived atrial models

Experimental findings have shown that atrial fibrosis results in changes that promote reentry (*Xia et al., 2004*; *Litchenberg et al., 2000*), but the exact mechanism of this connection is not fully understood. Previous modeling studies have linked RD localization to specific spatial patterns of fibrotic remodeling in AFib (*Zahid et al., 2016a*; *Hakim et al., 2018*). Recent clinical findings indicate that atrial fibrosis burden does not differ significantly between AFib and ESUS patients, and yet (by definition) ESUS patients do not demonstrate AFib at the time of stroke or during ambulatory monitoring (*Tandon et al., 2019*). Given the findings summarized above, a potential explanation is that, notwithstanding the fact that ESUS patients have substantial fibrosis, the particular *spatial distribution* of fibrotic remodeling in their atria is not conducive to arrhythmia perpetuation. Our findings suggest that this is likely not the case. From the standpoint of computational models derived from patient LGE-MRI scans, fibrotic substrate in individuals with ESUS is indistinguishable from that in patients with AFib in terms of the fundamental capacity to sustain RDs. For inducible models from both cohorts, we found that high fibrosis models were more likely to exhibit RDs irrespective of whether they corresponded to ESUS or AFib patients. This is in agreement with prior computational studies, which found the same general association (*Zahid et al., 2016a*; *Boyle et al., 2019*), and is consistent with clinical understanding, wherein higher fibrosis burden is associated with poor outcomes in AFib ablation procedures (*Marrouche et al., 2014*). Notably, we did observe several cases in which models defied inducibility expectations based on fibrosis alone. Such models exhibited RDs despite low fibrosis or were non-inducible despite high fibrosis. This observation confirms that, as observed previously in analogous AFib modeling studies (*Zahid et al., 2016a*; *Boyle et al., 2019*), assessment of raw fibrosis burden LGE-MRI scans alone is insufficient to fully characterize arrhythmogenic capacity of potentially pro-arrhythmic substrate in ESUS. In the present study, this is confirmed by our region-wise analysis of fibrosis spatial pattern (i.e., density, entropy), which is discussed below.

Consistent with the goal of this research to understand the contribution of fibrotic substrate to potential RD formation in ESUS, we purposefully excluded arrhythmias perpetuated by other mechanisms from our study design (e.g., self-sustaining activity driven solely by focal sources). This allows us to study the fibrotic substrate in the absence of all other confounding factors. Additionally, our analysis of abnormal steady-state depolarization in LA models suggests the potential for fibrosis itself to serve as a source of arrhythmia triggers is no greater in AFib than in ESUS models. Potential contributions from the right atrium (RA) were also excluded, since only the LA was segmented from LGE-MRI as part of the clinical workflow. Either of these factors may explain the absence of simulated arrhythmia in 23 of 45 AFib models, many of which had very little LA fibrosis (i.e., AFib in these individuals might have been predominantly focal in nature or sustained by RDs in the RA). This rate of inducibility is consistent with previous studies (e.g., 13 of 20 in computational models reconstructed from LGE-MRI scans of persistent AFib patients) (*Zahid et al., 2016a*), supporting the notion that LA fibrosis is associated with increased arrhythmia inducibility but fails to tell the whole story. Importantly, neither of these model constraints repudiates the central finding of our study, which suggests that there is no difference between ESUS and AFib patients in terms of the fundamental capacity of the fibrotic substrate to potentially harbor RDs.

## RD localization dynamics and morphology in patient-derived atrial models

As discussed above, our qualitative findings suggest that ESUS patients' fibrotic substrate is no different than that of AFib patients in terms of the capacity to sustain RDs per se. We performed additional analysis to assess whether specific consequences of fibrotic remodeling influenced any characteristics of RD inducibility in different ways for simulations in models corresponding to ESUS vs. AFib patients. First, we found that there was no significant difference in global fibrosis burden between inducible AFib and ESUS models. Thus, our computational modeling suggests that intrinsic pro-arrhythmic traits of ESUS and AFib fibrotic substrate are indistinguishable. Given this result, one potential alternative explanation for the lack of arrhythmia in ESUS patients is a lack of suitable triggers, despite an abundance of fibrotic substrate on par with that observed in AFib. The plausibility of this explanation is strengthened by the fact that the pacing sites from which episodes of reentry

were induced in our study were based on common AFib trigger sites as identified in a recent clinical study (*Santangeli and Marchlinski, 2017*); this is in contrast to previous modeling studies, which simulated triggered activity from evenly-distributed atrial sites (*Boyle et al., 2019*). Nevertheless, the premise of this aspect of our work was not to assess an absence of triggers, but instead ask the following: if the atria of these ESUS patients were subjected to the same type of triggered activity known to occur in typical AFib patients, is it possible the result be would sustained arrhythmia? In more than half of the cohort (24/45), our analysis suggests that the answer is yes. This new hypothesis could be validated in future studies by designing clinical protocols that systematically monitor ESUS patients with different levels of fibrosis for potential AFib triggers via electrocardiographic readouts.

Further analysis was performed to probe potential differences between ESUS and AFib simulations that went beyond consideration of RD inducibility as a binary variable. Specifically, we found no difference in the number of unique model-predicted RDs or the number of pacing sites that induced RDs between ESUS and AFib. Instead, these variables were highly correlated with LA fibrosis burden, which is consistent with the concept that high fibrosis models are more susceptible to pacing-induced RDs. This finding further substantiates our principal claim that no significant differences exist between the detected fibrotic substrate in ESUS and AFib, in that it holds true for the substrate's capacity to sustain reentry and its susceptibility to triggered activity. The general implication is that in the presence of simulated triggered activity, both of these characteristics are closely linked to global fibrosis burden.

RDs identified by non-invasive electrocardiographic imaging (ECGI) and in silico phase singularity identification have been shown to co-localize with fibrosis boundary zones identified by LGE-MRI (*Zahid et al., 2016a*; *Roney et al., 2016*; *Boyle et al., 2018a*; *Cochet et al., 2018*). RD localization dynamics in this study were consistent with these findings, as illustrated by representative LA fibrotic tissue distributions and corresponding RDs in *Figure 4*. RD morphology in this study corroborated previous findings – arrhythmias were perpetuated by one RD at a time, with activity in the periphery including conduction block, transient reentry, and wavefront collision (*Zahid et al., 2016a*; *Boyle et al., 2019*).

## Insights from analysis of RD inducibility and localization by LA region

Further to the macroscopic analysis discussed above, our region-by-region analysis of RD inducibility and localization showed that spatial properties of the fibrotic substrate between AFib and ESUS models are not intrinsically different. However, this analysis yielded several noteworthy findings that suggest subtle distinctions in fibrosis pattern may exist between the two groups. The most striking difference exists for the posterior wall region, where the IdS score was $\approx 2\times$ higher in AFib compared to ESUS models. Thus, even in cases where posterior LA wall ectopic excitations occur in ESUS patients, our simulations suggest that they could be up to 50% less likely to engage the fibrotic substrate and initiate sustained reentry compared to the same activity in AFib patients. This possibility does not contradict our hypothesis that the lack of arrhythmia in ESUS is due to a dearth of triggers; rather, it is a complementary corollary that can be put to the test in future clinical and computational analysis.

While understanding of RD localization dynamics in AFib remains limited, evidence from ECGI mapping indicates that reentrant activity occurs most frequently in the PV and posterior wall regions (*Boyle et al., 2018a*; *Haissaguerre et al., 2014*; *Tanaka et al., 2007*). Our findings are consistent with these data for AFib but not ESUS models. Many (41.1%) AFib model RDs localized to the LPV region. In contrast, in the ESUS population the RD localization hotspot was the region comprising the anterior wall and LAA (37%). The importance of this finding is unclear, as tendencies toward reentrant activity in particular LA areas have not been meaningfully correlated to clinical arrhythmia properties and are potentially subject to changes in conduction velocity or action potential duration (*Deng et al., 2017*). However, it provides a path for future validation studies: if incident AFib in patients who previously presented with ESUS can be characterized by intracardiac mapping, the hypothesis that RDs localize preferentially to the anterior wall can be tested.

Finally, our analysis of fibrosis spatial patterns revealed that the proportion of tissue with the propensity to harbor RDs (as established via machine learning in prior work *Zahid et al., 2016a*) was higher in the LPV region for AFib compared to ESUS models. While we acknowledge that a more fulsome analysis will be required to draw comprehensive conclusions on this subject, we note that our

results are consistent with the most prominent observed regional variabilities in RD localization and inducibility between the two model groups. In contrast, ESUS patients trended toward a higher percentage of pro-RD tissue in all anatomical areas except the LPVs. Thus, despite some interesting and potentially consequential differences in regional distribution of potentially pro-arrhythmic fibrosis, our overarching conclusion remains unchanged: our models suggest that if the ESUS substrate were subjected to suitable triggered activity, it could sustain the same types of RDs as those that contribute to AFib perpetuation.

## Limitations

In this study, atrial tissue is modeled as a bilayer to drastically reduce computational load. Previous studies have used this modeling framework (*Roney et al., 2016*; *Labarthe et al., 2014*) to represent human atria effectively, but the framework remains a simplification compared to volumetric 3D models. Moreover, clinical-grade MRI resolution limits our ability to detect fine details in anatomical structure and spatial distribution of potentially arrhythmogenic substrate, for instance slow-conducting tracks of fibrotic atrial tissue that could underlie microreentrant circuits (*Hansen et al., 2018*). While these models are patient-specific in terms of LA anatomy and each individual's unique pattern of fibrotic remodeling, they do not incorporate inter-patient variability in conduction velocity (CV) and electrophysiological properties such as ion channel expression. Nevertheless, our previous analysis indicates that this representation of atrial architecture with generic 'average AFib' electrophysiology is appropriate for use in patient-derived modeling (*Hakim et al., 2018*; *Deng et al., 2017*).

As in previous studies (*Ali et al., 2019*; *Shade et al., 2020*), our models do not differentiate between cell- or tissue-scale properties of atrial electrophysiology between patients with paroxysmal and persistent forms of AFib. Likewise, our approach to characterizing potential arrhythmia propensity in ESUS patients assumes cell- and tissue-scale remodeling based on experimental and clinical data from the AFib milieu. Although this is relevant as a limitation and must be considered when interpreting our results, this aspect of our approach is also one of the major advantages of the modeling and simulation methodology. Specifically, it allows us to assess whether there are any relevant differences in the spatial pattern of fibrotic remodeling between ESUS and AFib patients in the absence of other potentially confounding variables. A related limitation is that patients with stroke were excluded from the AFib cohort. This was because stroke etiology in the database from which AFib patients were drawn was not explicitly adjudicated to be cardioembolic, other ischemic such as atherosclerotic, or hemorrhagic. Therefore, we would not be able to draw reliable conclusions regarding the role of fibrosis in stroke in this population.

Finally, the mechanism of stroke in ESUS patients may be independent of AFib. Decreased atrial function due to atrial fibrosis may contribute to reduced hemodynamic efficacy and thrombus formation in the absence of AFib. Currently, secondary stroke prophylaxis is dependent on detecting AFib and predicting, through computational modeling, which atria are more prone to manifest AFib may be of clinical value. Another future research direction that could prove highly fruitful would be to create multi-scale, multi-physics image-based models of the fibrotic atria to assess each individual's risk of clot formation in a patient-specific manner (*Boyle et al., 2021a*).

## Conclusions

Simulations suggest that the pro-arrhythmic properties of fibrotic substrate in ESUS and AFib patients are indistinguishable. Our results show that fibrotic remodeling in ESUS patients has the theoretical capacity to sustain reentry when subjected to common AFib triggers. Thus, we conclude that fibrotic substrate conducive to perpetuating reentry may exist in up to half of ESUS patients. As individuals studied in this cohort present with incident AFib over the next few years, we will be able to put this hypothesis to the test. Our findings also support the notion that the lack of AFib in this population may be attributable to a lack of suitable arrhythmic triggers, but further research is needed to fully justify this claim. While the existence of pre-clinical substrate is correlated with a higher global proportion of fibrotic tissue, many ESUS cases defied these expectations, suggesting that fibrosis burden alone is insufficient for predicting pre-clinical AFib substrate. This conclusion justifies the use of computational simulations to probe beyond the fibrosis as imaged. Overall, these results provide novel insights into the role of atrial fibrotic remodeling as a critical nexus between the otherwise distinct manifestations of AFib and ESUS.

## Materials and methods

### Patient population

Patients were recruited to undergo cardiac LGE-MRI from the University of Washington (Seattle, WA) and Klinikum Coburg (Coburg, Germany) between July 2016 and June 2019. This study was approved by the Institutional Review Board (IRB) of the University of Washington (UW) and the Ethik-kommission der Bayerischen Ländesärztekammer München, Bayern, Deutschland; all participants provided written informed consent. Patients with ESUS met published diagnostic criteria (*Hart et al., 2017*). Patients with paroxysmal (27/45, 60%) or persistent AFib and without stroke were recruited from the UW Cardiac Arrhythmia Data Repository, an IRB-approved database for arrhythmia patients. Exclusion criteria for AFib patients included those who had undergone LA catheter ablation before MRI and those with only atrial flutter. Patients with cardiac implantable electronic devices, severe claustrophobia, renal dysfunction, and other contraindications to MRI or gadolinium-based contrast were excluded.

### MRI acquisition

Cardiac LGE-MRI was obtained on all participants to quantify the extent of LA fibrosis using previously described protocols (*Marrouche et al., 2014*). Scans were performed on Philips Ingenia and Siemens Avanto clinical scanners, 15–25 min after contrast injection, using a three-dimensional inversion-recovery, respiration-navigated, ECG-gated, gradient echo pulse sequence. Acquisition parameters included transverse imaging volume with a voxel size of 1.25 × 1.25 × 2.5 mm (reconstructed to 0.625 × 0.625 × 1.25 mm). Scan time was 5–10 min dependent on respiration and heart rate. Fat saturation sequences were used to suppress signal from fatty tissue.

### Reconstruction of 3D patient-derived atrial models from LGE-MRI

Geometric models were reconstructed from LGE-MRI, and the relative extent of fibrosis in the LA was quantified via an adaptive histogram thresholding algorithm (*Jadidi et al., 2013*). Clinical-grade meshes (i.e., coarse discretization) produced by Merisight Inc (Salt Lake City, UT) were resampled with a target resolution of 200 μm using an automated process based on gmsh (*Geuzaine and Remacle, 2009*). Each LA model was represented as a bilayer comprising of nested endocardial and epicardial shells (*Labarthe et al., 2014*), linked at every point by linear connections ($\sigma$ = 0.8 S m$^{-1}$) (*Figure 1A*). LA bilayer models were generated by slightly inflating the single-surface mesh to form the epicardial surface (i.e., duplicating endocardial points then moving them outward by 100 μm along the surface normal vector), then connecting the nested shells by attaching linear elements between corresponding nodes. In each patient-derived model, realistic myocardial fiber orientations were mapped from an atlas geometry (*Labarthe et al., 2014*) using the universal atrial coordinates (UAC) approach. Briefly, this process assigned epicardial and endocardial fibers from a previously published bilayer model to the target atrial geometry (*Figure 1B*; *Roney et al., 2019a*; *Roney et al., 2021*). In all finite-element LA meshes, the average element edge length was ≈188 μm and the number of nodes ranged from ≈600,000 to ≈1.4 million, depending on LA size. This mesh resolution is consistent with previously established benchmarks for minimizing numerical error due to spatial discretization in simulations of cardiac wavefront propagation (*Niederer et al., 2011*; *Boyle et al., 2021b*).

### Modeling of atrial electrophysiology in fibrotic and non-fibrotic regions

Our methodology for computational modeling at the cell and tissue scale of the fibrotic and non-fibrotic atrial electrophysiology can be found in previously published papers (*Zahid et al., 2016a*; *Boyle et al., 2018a*; *Boyle et al., 2018b*). Briefly, in non-fibrotic regions, a human atrial action potential model (*Courtemanche et al., 1998*) was used to represent membrane kinetics, including parameter modifications to fit clinical monophasic action potential recordings from AFib patients ($I_{Kur}$, $I_{to}$, and $I_{CaL}$ decreased by 50%, 50%, and 70%, respectively) (*Zahid et al., 2016a*; *Krummen et al., 2012*). At the tissue scale, conductivity tensor values in non-fibrotic tissue (longitudinal: $\sigma_L$ = 0.409 S m$^{-1}$; transverse: $\sigma_T$ = 0.0820 S m$^{-1}$) were calibrated to obtain effective CV values of 71.49 cm s$^{-1}$ and 37.14 cm s$^{-1}$ (longitudinal and transverse). These conductivities were chosen as a compromise between CV values measured in patients induced AFib (61 ± 6 cm s$^{-1}$)

(*Konings et al., 1994*), clinically mapped patients with AFib or atrial flutter (median: 60 cm s$^{-1}$; inter-quartile range: 22 cm s$^{-1}$) (*Verma et al., 2018*), and simulations calibrated to match intracardiac mapping data from an individual with paroxysmal AFib (median: 143 cm s$^{-1}$ longitudinal, 94 cm s$^{-1}$ transverse) (*Roney et al., 2019b*). In fibrotic regions, modifications to the AFib-like action potential model ($I_{CaL}$, $I_{Na}$, and $I_{K1}$ decreased by 50%, 40%, and 50%, respectively) were implemented as in prior studies (*Zahid et al., 2016a*; *Zahid et al., 2016b*), resulting in a 15.4% increase in action potential duration and a 49.6% decrease in upstroke velocity. These changes represented the effect of elevated transforming growth factor-β1, a key component of the fibrogenic signaling pathway. As in previous studies (*Zahid et al., 2016a*; *Boyle et al., 2018a*; *Boyle et al., 2018b*), tissue-scale effects of interstitial fibrosis and gap junction remodeling were represented by reducing overall conductivity and exaggerating the anisotropy ratio ($\sigma_L$:$\sigma_T$) from 5:1 to 8:1 ($\sigma_L = 0.177$ S m$^{-1}$; $\sigma_T = 0.0221$ S m$^{-1}$).

## Simulation of electrical activity and numerical aspects

Electrical propagation in bilayer LA models was simulated by solving the monodomain equation using the finite-element method. This system was coupled with ordinary differential and algebraic equations representing myocyte membrane dynamics at each node in the mesh, as described in the prior section. All simulations were executed on the Hyak supercomputer system at the University of Washington using the openCARP software package (*Vigmond et al., 2003*; *Vigmond et al., 2008*), which is available for academic use (see https://openCarp.org). The compute time required to complete each unique simulation ranged from 1 to 10 hr. The total CPU time for all simulations conducted in all models was 13.4 years.

## Induction and analysis of reentrant atrial arrhythmias

Simulations were performed to assess the pro-arrhythmic propensity of the fibrotic substrate in each patient-derived model. Arrhythmia induction via rapid pacing was attempted from 15 pacing sites derived from AFib trigger sites (*Figure 1C*, see caption for detailed anatomical site descriptions) (*Santangeli and Marchlinski, 2017*). Clinically relevant AFib trigger sites were chosen over a random pacing schematic to specifically capture RDs that arise from locations demonstrated to induce AFib. As in previous publications, a clinically relevant pacing sequence of 12 electrical stimuli was delivered at each of the 15 locations (*Zahid et al., 2016a*; *Zahid et al., 2016b*). Individual cell-scale ionic models were paced to limit cycle at a rate basic cycle length of 500 ms. The electrical stimulus consisted of two initial pulses with a coupling interval of 300 ms, followed by pulses ramping down to 200 ms in 20 ms intervals. After the delivery of the final stimulus, simulations were monitored for self-sustaining electrical wavefront propagation. For all cases in which activity persisted for at least 5000 ms post-pacing, we applied further analysis to determine whether the cause was an induced RD or macroscopic reentry (i.e., continuous repetitive, self-sustaining activation propagating around a non-conductive obstacle such as the mitral valve or pulmonary vein(s)), which we consider flutter-like reentry. Instances of macroscopic reentry were excluded from further analysis.

For each AFib-inducible simulation, we documented whether each pacing site induced reentry and analyzed patterns of RD localization. Unique RD morphologies in each patient-derived model were classified as belonging to one of five anatomical regions (*Figure 1D*), which were delineated automatically in a process summarized in *Figure 1—figure supplement 1*. First, the LA was subdivided into three broad anatomical areas (region 1: LA floor, 2: posterior wall; 3: anterior wall including LAA) using standardized cutoff values in the UAC space (*Roney et al., 2019a*). Second, the left and right PV areas (regions 4 and 5, respectively) were established using a region-growing approach such that each accounted for 15% of the total LA surface area. We then defined region-wise inducibility scores (IdS) across all models in a particular group (ESUS or AFib) as the proportion of pacing sites within a given region from which rapid pacing resulted in initiation of an RD. For example, since the LPV region contains four pacing sites (anterior/posterior LSPV/LIPV), the corresponding ESUS IdS value would be derived by summing the number of instances in which pacing from those locations induced RD across inducible ESUS models then dividing by four. This ensured our ability to assess spatial heterogeneity of sensitivity to triggered activity in a manner that was unbiased to the relative abundance of pacing sites in some LA regions.

## Quantitative analysis of fibrosis spatial pattern and potentially pro-ectopic effects of fibrosis

Fibrotic tissue areas with both high local fibrosis density (FD) and high local fibrosis entropy (FE) have increased propensity for RD localization, as shown in computational models (*Roney et al., 2016*) and intracardiac mapping of patients with persistent AFib (*Cochet et al., 2018*). We used a constraint equation derived by support vector machine-based classification to delineate such pro-RD regions. As in prior studies (*Zahid et al., 2016a*; *Ali et al., 2019*), the classification polynomial was as follows: $0.4096\ FD^2 + 3.28(FD)(FE) - 0.1036\ FE^2 - 0.7112(FD) - FE + 0.0429$. Maps of pro-RD tissue were subdivided into the five LA regions, as described above, and region-specific burdens of pro-RD fibrosis pattern were calculated.

In some situations, fibrosis itself can develop the capacity to generate ectopic triggers of arrhythmia (*Gouvêa de Barros et al., 2015*; *Alonso et al., 2016*). Although the cell-scale models used in this study do not undergo early or delayed afterdepolarizations due to simplified intracellular calcium handling, fibrotic regions in our models do have a potentially pro-ectopic higher resting membrane voltage ($V_m$) compared to non-fibrotic regions due to reduced ion channel expression levels ($I_{CaL}$, $I_{Na}$, and $I_{K1}$). Thus, as a surrogate measure of potential intrinsic pro-trigger capacity in fibrotic tissue, we characterized the extent of tissue in ESUS and AFib models in which the fibrosis pattern resulted in abnormal depolarization. To do this, we allowed all 90 models to reach a quasi-equilibrium state (1000 ms in the absence of pacing). We aggregated resting $V_m$ values across all models and identified the 95th percentile as the threshold for delineation of fibrosis-induced abnormal depolarization (–75.3 mV). Then, we calculated the proportion of tissue in each model with an equilibrium $V_m$ above that threshold.

## Statistical analysis

LA models for ESUS and AFib patients were divided into quartiles based on the extent of fibrotic remodeling as measured by LGE-MRI. Continuous variables were compared pairwise between groups using Wilcoxon rank-sum tests and were reported as mean ± standard deviation. Confidence intervals were calculated as the interval for the true difference in mean with 95% certainty. Categorical variables were compared using a $\chi^2$ test. After classifying unique RDs and number of pacing sites that induced reentry, correlation with fibrosis was assessed with logistic regression. Statistical significance was established at two-tailed $p \leq 0.05$. All statistical analysis was performed using R (*R Development Core Team, 2019*).

## Acknowledgements

We would like to thank Dr. Colleen Clancy, Dr. Axel Loewe, and our anonymous peer reviewer for extremely helpful and constructive feedback on our manuscript. The funders had no role in study design, data collection and interpretation, or the decision to submit the work for publication.

## Additional information

### Funding

| Funder | Grant reference number | Author |
| --- | --- | --- |
| National Institutes of Health | T32-EB001650 | Savannah F Bifulco |
| Achievement Rewards for College Scientists Foundation | | Savannah F Bifulco |
| Medical Research Council | MR/S015086/1 | Caroline H Roney |
| National Institutes of Health | R01-HL152256 | Steven A Niederer |
| H2020 European Research Council | PREDICT-HF (864055) | Steven A Niederer |
| British Heart Foundation | RG/20/4/34803 | Steven A Niederer |
| Engineering and Physical Sciences Research Council | EP/P01268X/1 | Steven A Niederer |

| | | |
|---|---|---|
| Wellcome Trust | 203148/Z/16/Z | Steven A Niederer |
| National Institutes of Health | NIH 5-U01-NS095869 | David Tirschwell<br>WT Longstreth |
| University of Washington | John L. Locke Charitable Trust fund | Nazem Akoum |

The funders had no role in study design, data collection and interpretation, or the decision to submit the work for publication.

## Author contributions

Savannah F Bifulco, Conceptualization, Formal analysis, Validation, Investigation, Visualization, Methodology, Writing - original draft, Writing - review and editing; Griffin D Scott, Formal analysis, Validation, Writing - review and editing; Sakher Sarairah, Zeinab Birjandian, Data curation, Formal analysis, Validation, Visualization, Writing - review and editing; Caroline H Roney, Software, Validation, Methodology, Writing - review and editing; Steven A Niederer, Software, Methodology, Writing - review and editing; Christian Mahnkopf, Peter Kuhnlein, Marcel Mitlacher, David Tirschwell, WT Longstreth, Formal analysis, Validation, Investigation, Writing - review and editing; Nazem Akoum, Conceptualization, Resources, Data curation, Supervision, Investigation, Writing - original draft, Project administration, Writing - review and editing; Patrick M Boyle, Conceptualization, Resources, Data curation, Software, Formal analysis, Supervision, Funding acquisition, Validation, Investigation, Visualization, Methodology, Writing - original draft, Project administration, Writing - review and editing

## Author ORCIDs

Savannah F Bifulco (iD) https://orcid.org/0000-0001-6679-6716
Caroline H Roney (iD) https://orcid.org/0000-0001-6809-0928
Steven A Niederer (iD) https://orcid.org/0000-0002-4612-6982
Nazem Akoum (iD) http://orcid.org/0000-0002-2001-6806
Patrick M Boyle (iD) https://orcid.org/0000-0001-9048-1239

## Ethics

Human subjects: This study was approved by the Institutional Review Board (IRB) of the University of Washington (UW) and the Ethikkommission der Bayerischen Ländesärztekammer München, Bayern, Deutschland; all participants provided written informed consent. Associated reference numbers: IRB5350 for ESUS patients; IRB8763 for AFib patients.

## Decision letter and Author response

Decision letter https://doi.org/10.7554/eLife.64213.sa1
Author response https://doi.org/10.7554/eLife.64213.sa2

# Additional files

## Supplementary files

• Transparent reporting form

## Data availability

Where possible (Figs. 2, 3, 5, 6), raw numerical data underlying figures are available via figshare: https://doi.org/10.6084/m9.figshare.14348042. Patient-derived data related to this article, including processed versions thereof, are not publicly available out of respect for the privacy of the patients involved. Interested parties wishing to obtain these data for non-commercial reuse should contact the co-corresponding authors via email. Upon all reasonable requests for access to these data, the co-corresponding authors will work to pursue negotiation of a Data Transfer and Use Agreement with the requesting party; administrators at the requesting party's institution, the University of Washington, and Klinikum Coburg; and relevant Institutional Review Boards at all the latter institutions.

Source files for a complete example of computational modeling and simulation of the fibrotic atria, using publicly available data sets and software tools only, can be found via the following permanent link: https://doi.org/10.6084/m9.figshare.14347979. Documentation provided with this example includes instructions on the use of the openCARP cardiac electrophysiology simulator and the meshalyzer visualization software (both available via https://opencarp.org/) to precisely reproduce the computational protocol applied to patient-specific left atria models in this study.

The following dataset was generated:

| Author(s) | Year | Dataset title | Dataset URL | Database and Identifier |
|---|---|---|---|---|
| Bifulco SF, Scott GD, Sarairah S, Birjandian Z, Roney CH, Niederer SA, Mahnkopf C, Kuhnlein P, Mitlacher M, Tirschwell D, Longstreth WT, Akoum N, Boyle PM | 2021 | Source Data for Study by Bifulco et al. | http://doi.org/10.6084/m9.figshare.14348042 | figshare, 10.6084/m9.figshare.14348042 |

The following previously published dataset was used:

| Author(s) | Year | Dataset title | Dataset URL | Database and Identifier |
|---|---|---|---|---|
| Boyle PM | 2021 | Computational Modeling & Simulation of Atria with Fibrotic Remodeling - Example | https://doi.org/10.6084/m9.figshare.14347979 | figshare, 10.6084/m9.figshare.14347979 |

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
