## [Decision Letter]

**Acceptance summary:**

The authors studied differences in the spatial distribution of atrial fibrosis comparing patients suffering embolic stroke of undetermined source and patients with atrial fibrillation without stroke history. Using MRI-based computational modeling the authors show that in patients with inducible atrial fibrillation fibrosis was slightly higher but that fibrosis was similar whether or not there was a history of stroke. These findings suggest common trigger mechanisms underlying inducible atrial fibrillation, irrespective of stroke history. These novel findings are of great interest for clinicians and research scientists working on the pathogenesis of atrial fibrillation.

**Decision letter after peer review:**

Thank you for submitting your article "Computational modeling identifies embolic stroke of undetermined source patients with potential arrhythmic substrate" for consideration by *eLife*. Your article has been reviewed by 3 peer reviewers, one of whom is a member of our Board of Reviewing Editors, and the evaluation has been overseen by a Senior Editor. The following individuals involved in review of your submission have agreed to reveal their identity: Axel Loewe (Reviewer #2); Colleen Clancy (Reviewer #3).

Essential revisions:

1. The authors could go farther in describing why this is surprising. Generally, severe fibrosis is thought to potentially serve as a means or mechanism for pro-arrhythmic triggers. This is because damage to cardiac tissue typically results in calcium dysregulation. When calcium overload occurs in isolated fibrotic tissue areas, or depolarization of the resting membrane potential due to localized ischemia allows for ectopic pacemaking, we might expect that the diseased/fibrotic tissue is itself the source of arrhythmia generation. The novel finding here is that this notion may be a simplification, and the sources of arrhythmia generation may be more complex and may need to come from outside the areas of fibrosis.

2. Can the authors provide more explanation for the fiber orientation maps shown in figure 1B? Shouldn't the topology and surface area of the endocardial fiber area be substantially different than the epicardial surface? It looks as though the endocardial fiber orientation was mapped onto the epicardial surface. Some clarity around this would be helpful.

3. In the aggregated data and figure 3, where the authors demonstrate that fibrosis burdens for inducible and non-inducible models or significantly different, can they say why? Does this relate directly to the extent of fibrosis? Is this essentially the same data that's presented in figure 2, that describes the relationship between the fibrotic burden and inducibility?

4. The authors state that their simulations suggest that triggered activity in the posterior wall maybe more likely to initiate reentrant arrhythmia in AFib patients compared to ESUS patients. Is this correlated in any way to the pattern of fibrosis observed in the two different groups? It would be nice to get more mechanistic insight here. In the discussion the authors describe the spatial distribution of fibrotic remodeling as a potential factor leading to the differences between the two groups. Isn't it possible to examine and quantify the special distribution? This feels like a prime application for machine learning.

5. L280: "This observation suggests that even in the cases when common potential LA triggers, such as (i.e., PV or posterior wall ectopy) do occur in ESUS patients, they are less likely to engage the fibrotic substrate and initiate sustained reentry compared to the same activity in AFib patients." This statement should be revised to make clearer that it pertains only to the PV regions and the posterior LA wall.

6. L399: Can the authors elaborate on the bilayer model? The standard monodomain would either be solved on a 2D-surface mesh or a volumetric model, the generalization to a bilayer model has been reported before but should be described in more detail / referred to, to enable reproduction.

7. Because the authors mentioned that there is a marked difference in LA surface area and BMI between the groups, it would also be better if the authors could actually analyze this aspect further, such as actually normalizing the LA surface area to body surface area, to confirm whether this factor could affect their findings.

8. While the discussion does a good job in general, it could be made more concise by focusing on the discussion of results and shortening their summary.

---

## [Author Response]

Essential revisions:1. The authors could go farther in describing why this is surprising. Generally, severe fibrosis is thought to potentially serve as a means or mechanism for pro-arrhythmic triggers. This is because damage to cardiac tissue typically results in calcium dysregulation. When calcium overload occurs in isolated fibrotic tissue areas, or depolarization of the resting membrane potential due to localized ischemia allows for ectopic pacemaking, we might expect that the diseased/fibrotic tissue is itself the source of arrhythmia generation. The novel finding here is that this notion may be a simplification, and the sources of arrhythmia generation may be more complex and may need to come from outside the areas of fibrosis.

Broadly speaking, we agree with the review team. Sources of arrhythmia generation may come from areas outside of fibrotic tissue. Our simulations seek to represent the capacity of each individual’s unique fibrotic substrate to sustain reentrant drivers of arrhythmia; the explicit exclusion of other potentially arrhythmogenic factors is a *key feature* of our design, in that it allows us to study the fibrotic substrate in the absence of all other confounding factors. We have reinforced this point via a brief addition to the Discussion section.

In response to the review team’s specific suggestion, we ran additional simulations to rule out the possibility that fibrotic tissue might serve as a source of arrythmia generation (in AFib but not ESUS models) via ectopic pacemaking due to depolarized resting membrane potential. Fibrotic regions in our simulations have an intrinsically higher resting potential compared to non-fibrotic regions due to changes in ion channel expression (reduced I_CaL_, I_Na_, I_K1_). Our new results show that there is no significant difference in the extent of significantly depolarized tissue between models representing the atria of ESUS and AFib patients (P=0.32). We have added this analysis to Figure 3F. This supports the review team’s sentiment that “trigger-centric” conventional wisdom regarding links between fibrosis and arrhythmogenesis is an over-simplification; this point has also been emphasized in our Discussion section.

2. Can the authors provide more explanation for the fiber orientation maps shown in figure 1B? Shouldn't the topology and surface area of the endocardial fiber area be substantially different than the epicardial surface? It looks as though the endocardial fiber orientation was mapped onto the epicardial surface. Some clarity around this would be helpful.

We are extremely grateful to Dr. Loewe for this observation. A close inspection of these images led us to an error in the directionality of the fibers in a small subset of our patient specific models. As a result, we fixed the defect in our model reconstruction process and repeated all simulations and analysis for the affected cases. We revised images in Figure 1B to highlight the major differences between endocardial and epicardial fiber patterns, which are more prominent on the anterior wall compared to the posterior wall. These revisions did not change the major findings of our study (i.e., no changes in results of statistical analysis at the cohort scale); in fact, there is now even stronger evidence (i.e., more convincing P values) suggesting that pro-arrhythmic properties of fibrosis in ESUS and AFib are indistinguishable in these models. Nevertheless, as noted in our preamble to this response, there were subtle changes in summary data for ESUS and AFib cases throughout the study (Figures 2-4, original Figure 6, now renumbered as Figure 5) and one observation from our original study (dual reentrant driver feedback; original Figure 5 and Supplemental Video 1) were removed.

3. In the aggregated data and figure 3, where the authors demonstrate that fibrosis burdens for inducible and non-inducible models or significantly different, can they say why? Does this relate directly to the extent of fibrosis? Is this essentially the same data that's presented in figure 2, that describes the relationship between the fibrotic burden and inducibility?

Yes, both figures emphasize the observation that reentrant driver inducibility in these models is strongly associated with fibrosis burden. This is consistent with prior computational studies, which found the same general association (e.g., *Zahid et al. (2016) Cardiovasc Res). From the clinical standpoint, higher fibrosis burden is associated with poor outcomes in AFib ablation procedures (e.g., Marrouche et al. (2014) JAMA)*. However, it is known that fibrosis burden alone is not enough to completely predict reentrant driver inducibility. Prior simulation work has shown that area with a spatial fibrosis pattern that has high local density *and* entropy are most likely to harbor reentrant activity. We have added clarifying text to the manuscript to make this point more explicitly.

On the question of redundancy between figures: While there is significant overlap between the findings presented in these two figures, we believe there is value in showing both, as they present the same results in a different light. Figure 3A stresses the contrasts between groups; Figure 2B emphasizes association between inducibility and fibrosis, while also highlighting the small number of cases that defy this relationship (i.e., models that have regions with high fibrosis density and entropy despite low global fibrosis burden). Of note, the format for the latter graph parallels that of Figure 3 from *Tandon et al. (2019) Neurology.*

4. The authors state that their simulations suggest that triggered activity in the posterior wall maybe more likely to initiate reentrant arrhythmia in AFib patients compared to ESUS patients. Is this correlated in any way to the pattern of fibrosis observed in the two different groups? It would be nice to get more mechanistic insight here. In the discussion the authors describe the spatial distribution of fibrotic remodeling as a potential factor leading to the differences between the two groups. Isn't it possible to examine and quantify the special distribution? This feels like a prime application for machine learning.Please note, there are subtle differences between precise results discussed here and those originally presented due to the corrected method for atrial fiber mapping, as discussed above.

This is a great suggestion for deeper analysis. The data presented in our figure on reentrant driver inducibility and localization (originally Figure 6; now Figure 5), we found that there was no difference in region-wise spatial distribution of pacing sites that initiated arrhythmia (P=0.45); similarly, in the case of reentrant driver localization site distribution, there was no discernible regional trend (P=0.13). However, we did observe interesting regional trends (e.g., higher prevalence of localization sites in the LPV region for AFib models and the anterior wall region for ESUS models). Based on the review team’s suggestions, we performed new analysis (new Figure 6 and accompanying text in Results and Methods) to delve deeper into whether this might be due to region-wise trends in fibrosis spatial pattern. Indeed, we now present findings that confirm there is a higher extent of tissue with a pro-arrhythmic fibrosis pattern in one of areas discussed above (LPV in AFib). While we did not find a significant difference in the anterior wall of ESUS models, there was a trend towards higher pro-arrhythmic fibrosis patterns in that area. Implications of all the latter analysis points are noted in our revised Discussion.

5. L280: "This observation suggests that even in the cases when common potential LA triggers, such as (i.e., PV or posterior wall ectopy) do occur in ESUS patients, they are less likely to engage the fibrotic substrate and initiate sustained reentry compared to the same activity in AFib patients." This statement should be revised to make clearer that it pertains only to the PV regions and the posterior LA wall.

We agree that this is a necessary clarification. The text in question has been amended to be consistent with the conclusions that can be drawn from the analysis in the figures on reentrant driver inducibility/localization (old Figure 6, new Figure 5) as well as the new analysis on fibrosis density and entropy (new Figure 6). Please note that the specific regions in question have changed due to the aforementioned corrections to the model reconstruction pipeline.

6. L399: Can the authors elaborate on the bilayer model? The standard monodomain would either be solved on a 2D-surface mesh or a volumetric model, the generalization to a bilayer model has been reported before but should be described in more detail / referred to, to enable reproduction.

Our implementation of the bilayer model (Methods Section: Reconstruction of 3D patient derived atrial models from LGE-MRI) is generated by slightly inflating a surface 2D mesh (i.e., duplicating endocardial points then moving them outward by 100 µm along the surface normal vector at that location), then connecting the two surfaces by attaching linear elements between the corresponding nodes. With this formulation of the bilayer model, there are no alterations needed in the implementation of the openCARP solver. We have included a brief description and an additional reference to prior work with more detailed information.

7. Because the authors mentioned that there is a marked difference in LA surface area and BMI between the groups, it would also be better if the authors could actually analyze this aspect further, such as actually normalizing the LA surface area to body surface area, to confirm whether this factor could affect their findings.

We thank the review team for this astute comment. In fact, while the LA surface area and BMI values are provided in the interest of complete transparency, the preferred clinical metric for unbiases comparison of LA size between individuals is the left atrial volume index (i.e., ratio between LA volume and estimated body surface area). These data are already presented in Table 1 of our manuscript and show no statistically significant difference between ESUS and AFib cohorts. A deeper analysis of LA surface area and BMI values would be less clinically interpretable and is unlikely to provide meaningful insights. We have added clarifying text to this effect.

8. While the discussion does a good job in general, it could be made more concise by focusing on the discussion of results and shortening their summary.

The review team’s critique is duly noted and appreciated. Where possible, we have removed repetitive or extraneous text from the Discussion section. We do note that this part of our manuscript remains quite long due to the addition of several salient points raise by the review team, which we feel greatly improve the overall quality of the study.